# Molecular Composition and Biological Activity of a Novel Acetonitrile–Water Extract of Lens Culinaris Medik in Murine Native Cells and Cell Lines Exposed to Different Chemotherapeutics Using Mass Spectrometry

**DOI:** 10.3390/cells12040575

**Published:** 2023-02-10

**Authors:** Annamaria Di Turi, Marina Antonacci, Jacopo Raffaele Dibenedetto, Fatima Maqoud, Francesco Leonetti, Gerardo Centoducati, Nicola Colonna, Domenico Tricarico

**Affiliations:** 1Department of Pharmacy-Pharmaceutical Science, University of Bari “Aldo Moro”, Via Orabona 4, 70125 Bari, Italy; 2Department of Medicine Veterinary, University of Bari “Aldo Moro”, Str 62 to Casamassima, Valenzano, 70121 Bari, Italy; 3Terre di Altamura Srl, 70022 Altamura, Italy

**Keywords:** Lens culinaris, chemopreventive, cytotoxicity, primary murine cells, flavonoids, liquid chromatography–mass spectrometry, phenolic compounds, muscle fibers

## Abstract

We evaluated the effects of a new extract (70% acetonitrile, 2E0217022196DIPFARMTDA) of Lens culinaris Medik (Terre di Altamura SRL, Altamura BA) to prevent cytotoxic damage from cisplatin, staurosporine, irinotecan, doxorubicin, and the glucocorticoid dexamethasone. The acetonitrile–water extract (range 0.1–5 mg/mL) was obtained by extracting 10 g of lentil flour with 50 milliliters of the acetonitrile–water extraction mixture in a 70:30 ratio, first for 3 h and then overnight in a shaker at room temperature. The next day, the extract was filtered and passed through a Rotavapor to obtain only the aqueous component and eliminate that with acetonitrile, and then freeze-dried to finally have the powdered extract. In vitro experiments showed that the extract prevented the cytotoxic damage induced by cisplatin, irinotecan, and doxorubicin on HEK293 and SHSY5Y cell lines after 24–96 h. In murine osteoblasts after 24–72 h of incubation time, the extract was cytoprotective against all chemicals. The extract was effective against dexamethasone, leading to synergic cell proliferation in all cell types. In bone marrow cells, the extract is cytoprotective after 72 h against doxorubicin, staurosporine, and dexamethasone. Instead, on muscle fibers, the extract has a synergic effect with chemotherapeutics, increasing cytotoxicity induced by doxorubicin and staurosporine. LC-MS attested to the existence of several phenolic structures in the extract. The most abundant families of compounds were flavonoids (25.7%) and mellitic acid (18%). Thus, the development of this extract could be implemented in the area of research related to the chemoprevention of damage to renal, neuronal, bone marrow cells, and osteoblasts by chemotherapeutics; moreover, it could be used as a reinforcer of cytotoxic action of chemotherapeutics on muscle fibers.

## 1. Introduction

Legumes are an important component of the human diet because they are a significant source of plant protein, which is why many national guidelines place them at the base of the nutrition food pyramid. Ample scientific evidence recognizes these foods as fundamental in maintaining good health and preventing chronic diseases (cardiovascular diseases, neoplasms, respiratory diseases, diabetes, and other minor ones) responsible for many deaths in Italy. Since the renowned positive effects of legumes on health and sustainable agriculture systems are well known, interest in them has increased in recent years, and the FAO declared the year 2016 as the “International Year of Legumes.” Our study is focused on Lens culinaris, which is an annual native plant, known for its edible lens-shaped seed, which has one of the most important macro- and micronutrient compositions in plants [1]. It is an important resource of protein and provides the human body with both essential and non-essential amino acids. The main protein found in its composition is globulin, and it also contains albumin [2]. These seeds are relatively low in sodium and fat but high in potassium, which makes them convenient food for people who suffer from cardiovascular diseases or obesity. In addition, lentils contain a high quantity of polyphenols and bioactive food components, which are found to be in higher amounts than other common legumes such as peas, chickpeas, green beans, and peanuts [1]. Polyphenols and their derivatives are bioactive compounds found widely in foods of plant origin (fruits, vegetables, vegetables, legumes, cereals, seeds, spices, wine, tea, coffee, cocoa, and herbs) [3]. These natural organic substances have recently attracted much interest because they possess several biological activities that could explain their possible role in the prevention of chronic diseases. They are a large group of organic compounds and plant secondary metabolites. The polyphenolic composition of plants is highly variable, both qualitatively and quantitatively; some of them are ubiquitous, while others are specifically found in certain families or only in certain species. So far, about 8000 plant phenol structures have been described. Specifically, flavonoids are divided into six main subclasses: isoflavones, flavones, flavonols, anthocyanins, flavanones, and flavanols, which are usually associated with sugar, originating in glycosides [4]. Flavonols (e.g., quercetin, kaempferol, and myricetin) are the most abundant flavonoids in plant foods and are found mainly in leafy vegetables, apples, onions, and berries. Flavones (e.g., apigenin and luteolin) and anthocyanidins are present in smaller amounts in leafy vegetables and herbs. Flavonoids have numerous biological effects in cancer prevention: they are free radical scavengers, modulators of cell signaling and the cell cycle, antimutagenic, antiproliferative, and inhibit angiogenesis [4].

In our previous study [5], we evaluated the effects of a novel hydroalcoholic extract (70% ethanol, 1E030521DIPFARMTDA) from Lens culinaris to prevent cytotoxic damage from the same chemotherapeutic agents used also in this study. This extract prevented the cytotoxic damage induced by cisplatin and staurosporine on the renal cell line, and by cisplatin also on the neuronal cell line by clonogenic assay and DNA staining. The extract also prevented the reduction in cell proliferation induced by dexamethasone in murine osteoblasts, suggesting potential use in bone loss diseases. A UV/VIS spectroscopy analysis was carried out and three peaks were obtained at three different wavelengths, which correspond to groups of compounds such as flavonoids, proanthocyanins, and salicylates. At that point, it was not yet clear which of these groups of molecules composing the extract was responsible for these effects, or if there was a synergic mechanism.

Encouraged by the results of the previous study, we investigated in the present work a new extract (70% acetonitrile, 2E0217022196DIPFARMTDA) of Lens culinaris Medik against the cytotoxicity induced by different chemicals such as doxorubicin, cisplatin, irinotecan, staurosporine, and dexamethasone in cell lines and primary murine cell culture. This acetonitrile–water extract (range 0.1–5 mg/mL) prevents the cytotoxicity induced by irinotecan, cisplatin, and doxorubicin on HEK293 renal and SHSY5Y neuronal cell lines after 24–96 h of incubation time. In addition, the extract is sometimes effective with dexamethasone, leading to synergic cell proliferation. Moreover, this new extract shows cytoprotection of osteoblasts and bone marrow cells treated with chemotherapeutics. On muscle fibers it does not induce cytoprotection; in contrast, it has a synergic effect with chemotherapeutics, such as doxorubicin and staurosporine.

We also performed a comparative LC-MS analysis focusing on the composition and biological activity of the hydroalcoholic extract already investigated [5] and of the novel acetonitrile–water extract.

## 2. Materials and Methods

### 2.1. Preparation of Acetonitrile–Water Extract

Lens culinaris powder was obtained by finely grinding lentil seeds supplied by the company Terre di Altamura S.r.l. Bari, Italy, in a metal blade blender until a homogeneous flour of minimum particle size was obtained. From this, acetonitrile–water extract (70:30) was prepared. For the acetonitrile–water extract preparation, 10 g of lentil flour was extracted with 50 mL of the acetonitrile–water extraction mixture in a 70:30 ratio first for 3 h and then overnight (about 16 h) in a shaker at room temperature. The next day, the extract was filtered twice with paper filters (Whatman Filter papers, diameter 110 mm), passed through a Rotavapor to obtain only the aqueous component and eliminate the component with acetonitrile, and then freeze-dried to obtain the powdered extract. For subsequent tests, the extract was partly resuspended in PBS and stored at −20 °C and partly stored as a powder at −4 °C. The range of concentrations tested was from 0.01 mg/mL to 5 mg/mL at T = 37 °C for incubation periods from 3 to 96 h.

### 2.2. Drugs and Chemical Solution

The drugs and chemicals were purchased from Sigma (SIGMA Chemical Co., Milan, Italy). Stock solutions of the drugs we investigated were prepared by dissolving the compounds in dimethylsulfoxide (DMSO). The stock solution of staurosporine was prepared at a concentration of 5 mg/mL, which was diluted in DMEM at a 21.4 × 10^−6^ M concentration to obtain a final concentration of 2.14 × 10^−6^ M of staurosporine in the wells. The stock solution concentration of diazoxide was 118.6 × 10^−3^ M which was diluted in DMEM at a 250 × 10^−5^ concentration to give a final concentration of 250 × 10^−6^ M of diazoxide in the wells. The stock solutions concentrations of cisplatin and irinotecan were 0.5 M, diluted in DMEM at a 1 × 10^−3^ M concentration to obtain a final concentration of 1 × 10^−4^ M of cisplatin and irinotecan. The stock solution concentration of doxorubicin was 86.2 × 10^−3^ M, which was diluted in DMEM at a 100 × 10^−3^ M concentration to give a final concentration of 1 × 10^−6^ M of doxorubicin in the wells. For the dexamethasone, the stock solution concentration was 0.25 M, which was diluted in DMEM at a 2 × 10^−5^ M concentration to give a final concentration of 2 × 10^−6^ M (dexamethasone) in the wells. Microliter amounts of the stock solutions were then added to DMEM+ for crystal violet staining and direct fiber count by visual inspection.

### 2.3. Cell Culture

All cell lines and primary murine cells used were maintained in culture using Dulbecco’s Modified Eagle Medium (DMEM), to which 10% fetal bovine serum (FBS), 1% penicillin/streptomycin, and 1% L-glutamine were added. Cultures were maintained at 37 °C in a humidified environment containing 95% air and 5% CO_2_. Experiments were conducted on undifferentiated cells and primary cultures. The whole procedure was conducted under sterile conditions to prevent contamination by bacteria and molds. The resulting cells were cultured with DMEM enriched with 10% fetal bovine serum (FBS), 1% L-glutamine, and 1% antibiotics (penicillin–streptomycin) and placed in an incubator under standard conditions. Non-adherent cells were carefully removed after 3 h of incubation.

### 2.4. Crystal Violet Staining Test

Crystal violet assay to evaluate the efficacy of the extract was conducted following the protocol described below. Cells were plated in a multi-well plate at a density of 8 × 10^3^ cells/well. Cells were previously counted using the Scepter 2.0 counter (Merk Millipore Corporation, New York, NY, USA). After 24 h of incubation, the time required for the cells to adhere to the bottom of the plate, different concentrations of extract, previously solubilized in phosphate-buffered saline (PBS), which is harmless to cell cultures, were added. After the established incubation time (18–24–48–72 or 96 h), the medium was removed, the cells were fixed with 10% buffered formalin for 20 min at room temperature and then stained using a 1% *v*/*v* solution of Crystal Violet for 30 min. The plates were washed with distilled water to remove excess dye. Finally, acetic acid was added to elute the dye, and absorbance was read at 560 nm using the VictorTM spectrophotometer. Each experimental condition was evaluated in triplicate.

### 2.5. Primary Murine Cell Culture and Muscle Fibers Survival Evaluation

Tissues explanted from mice (WT/WT) were used to prepare native cells, and these were also used for other experimental purposes [6]. The various experimental protocols and animal care are regulated by European Directive 2010/36/EU of Animal Protection Used for Scientific Experiments and are approved by the Italian Minister of Health and also by the University of Bari Committee (prot. 8515-X/10, 30 January 2019). Osteoblasts and bone marrow cells were obtained from the murine tibia and femur [7]. The bones were dislocated, divided, washed gradually with dilutions of ethanol, and placed in a Petri dish in PBS. After that, we collected the bone marrow, and we used a 5 mL syringe to perfuse a sterile solution of PBS inside the medullary cavity [8]. Sterile conditions were practiced throughout the entire procedure to prevent contamination.

The effects (0–24 h) of the acetonitrile–water extract against staurosporine and doxorubicin were also tested on fiber morphology. Evaluation of morphological parameters of FDB fibers was performed by seeding fibers in the culture medium (DMEM supplemented with 10% fetal bovine serum, 1% L-glutamine, and 1% penicillin–streptomycin), at 37 °C. The isolated fibers were incubated in the culture medium for at least 30 min at 37 °C. Dead fibers were defined as cells that showed marked changes of ≥40% in morphological parameters such as length and diameter within 24 h, evaluated microscopically. The appearance of multiple protrusions on the sarcolemma preceded cell death. On isolated native fibers, death was expressed as the degree of mortality. The first degree of mortality indicated a mortality rate of 0–25%, the second degree indicated a mortality rate of 25–50%, the third degree of mortality indicated a mortality rate of 50–75%, and the fourth degree of mortality indicated a mortality rate of 75–100%. The FDB fibers were successfully used for in vitro experiments showing a high survival rate of >70% after 36 h of incubation time [5,6,9,10,11,12].

### 2.6. ScepterTM Cell Counter

The ScepterTM 2.0 (Merk Millipore Corporation, New York, NY, USA) cell counter is an automatic and portable cell counter that allows accurate and reliable cell counting in less than 30 s. To count cells, the instrument employs an impedance method. The Scepter can count cells through disposable sensors. These sensors consist of precision chambers with an electronic detection zone and integrated cell detection electrodes, which can discriminate cells based on their size and volume. The counting technology is based on the principle that the electronic resistance between two compartments filled with a solution of electrolyte changes when a particle, such as a cell, passes through a connecting pore between the two compartments.

The voltage increases in proportion to the resistance. Thus, each cell passage is related to an increase in resistance and a peak in voltage. The peaks of the same size are averaged in a histogram so that at the end of the analysis, the cytometer shows, in a single graph, the distribution of cells according to their size and volume and gives the exact number of cells. The Scepter can count approximately 1 × 10^4^ to 5 × 10^5^ cells, using sensors of two different diameter ranges: 40 μm sensors are used to count particles between 3 μm and 17 μm, while 60 μm sensors can count particles between 6 μm and 36 μm. For our experiments, we used 60 μm sensors. After turning on the Scepter, we attached the sensor to the base of the instrument with the electrode detection panel facing the front of the instrument. After attaching the sensor, we followed the instructions on the display. Thus, by dipping the sensor into the cell suspension solution with the device button down, and then releasing it, the Scepter draws the cell suspension into the sensor. The sample is loaded into the sensor and then through the detection zone, the size and volume of the cells are assessed. A total of 50 μL of the cell suspension was analyzed. At the end of the counting process, we removed and discarded the sensor, and the instrument provided the concentration (number of cells/mL) and histogram of cell diameter and size. The Scepter can store 70 histograms that can later be downloaded to a PC and analyzed using ScepterTM 2. Software Pro, Darmstadt, Germany. These histograms provide quantitative data on cell morphology that can also be used to assess the quality and health of cell cultures.

### 2.7. LC-MS Analysis (Liquid Chromatography–Mass Spectrometry)

Liquid chromatography–mass spectrometry (LC-MS) is an analytical technique based on the use of liquid chromatography together with mass spectrometry. The chromatograph separates the compounds in the sample while the mass spectrometer acts as a detector. In the analysis we performed, the instrument used was the Agilent 6530 accurate mass Q-TOF. This spectrometry analysis was performed in the negative (M-H), so we consider that compared to the actual *m*/*z* that is shown in the graph has one unit less. The compounds were identified using the program ChemSpider Advanced Search, Royal Soc. of Chemistry 2023, Cambridge, UK, and the properties of the substances were verified with the help of PubChem and PubMed databases.

### 2.8. Data Analysis and Statistics

The data obtained were collected and analyzed using Excel software (Microsoft Office 2010, Milano, Italy). The results are presented as the mean ± standard deviation. The significance of the results was assessed by performing a one-factor analysis of variance using Excel software (Microsoft Office 2010, Milano Italy), and statistical significance was assigned for values of *p* < 0.05 and variance ratio F > 1. Each experiment/well has its own specific calibration curve from which we calculated the cell numbers and the percentages of cell survival. The negative values for the percentage of cell survival were obtained when the absolute values of abs were very low, resulting in the negative values in the calculation of the cell numbers.

## 3. Results and Discussion

### 3.1. Evaluation of the Efficacy of Acetonitrile–H_2_O Extract in Preventing Doxorubicin and Dexamethasone-Induced Cytotoxicity on Osteoblasts

The extract shows cytoprotection against doxorubicin after all the incubation times in this cell type. More specifically, after 24 h of incubation, the cell survival rate is 112.43% if 2.5 mg/mL of extract is used with doxorubicin, versus 78.916% for doxorubicin alone. After an incubation time of 48 h, the cell survival rate is 117.439% using the extract against doxorubicin, versus 90% for doxorubicin alone. After 72 h of incubation, doxorubicin alone fully inhibits cell survival, but the survival cell rate increases to 53.6013% in the presence of the extract. After 48 and 72 h of incubation, 0.1 mg/mL extract shows mild cytoprotection, but less than the 2.5 mg/mL extract (Figure 1).

Dexamethasone does not show any cytotoxicity after incubation times of 24 and 48 h. Instead, after 72 h of incubation, this substance shows cytotoxicity, with a cell survival rate of 58.928%. In this condition, 2.5 mg/mL of extract has a strong cytoprotective effect against dexamethasone, with an increasing survival cell rate of 154.04%.

### 3.2. Evaluation of the Efficacy of Acetonitrile–H_2_O Extract in Preventing Irinotecan-Induced Cytotoxicity on Osteoblasts

Irinotecan failed to show any cytotoxicity after 24 h of incubation time but showed cytotoxicity after 72 h of incubation time. In this experimental condition, 2.5 mg/mL of extract has a cytoprotective effect, increasing the cell survival rate to 97.36% versus 86.52% when only irinotecan is used (Figure 2).

### 3.3. Evaluation of the Efficacy of Acetonitrile–H_2_O Extract in Preventing Staurosporine-Induced Cytotoxicity on Osteoblasts

Against staurosporine, 2.5 mg/mL of extract shows a cytoprotective effect both after 48 and 72 h of incubation. Specifically, after 48 h of incubation, the cell survival rate using only staurosporine is 77.86%. The cell survival rate will increase to 114.45% in the presence of 2.5 mg/mL extract; the extract shows cytoprotection also at 0.1 mg/mL concentration with a cell survival rate of 82.192%. After 72 h of incubation, staurosporine has a marked cytotoxic activity with a cell survival rate of only 10.25%. The coincubation of the cells with staurosporine and the 2.5 mg/mL extract increased the cell survival rate to 89.96% (Figure 3). The coincubation of diazoxide with staurosporine showed less cytoprotection than the extract in this cell type.

### 3.4. Evaluation of the Efficacy of Acetonitrile–H_2_O Extract in Preventing Doxorubicin-Induced Cytotoxicity on Bone Marrow Cells

Doxorubicin shows full cytotoxicity with no cell survival in this cellular type. After an incubation time of 48 h, the 2.5 mg/mL extract does not show a cytoprotective effect against doxorubicin; instead, in this condition, only diazoxide increases the cell survival rate to 61.15%. The strong cytotoxicity of this chemotherapeutic is highlighted after 72 h of incubation, where the cell survival rate is evaluated at −6.078%. After this incubation time, the extract at 2.5 mg/mL concentration showed a strong cytoprotective and cytoproliferative effect, with a 145.40% cell survival rate. However, after an incubation time of 96 h, 2.5 mg/mL of extract gives no more cytoprotection; instead, it increases doxorubicin cytotoxicity, with a cell survival rate of −61.68% versus −10.56% for doxorubicin used alone. However, 96 h is a very long incubation time, which may not have a clinical impact considering that chemotherapy exposure is no longer than 36–72 h of incubation (Figure 4).

### 3.5. Evaluation of the Efficacy of Acetonitrile–H_2_O Extract in Preventing Dexamethasone and Staurosporine-Induced Cytotoxicity on Bone Marrow Cells

After 48 h of incubation, the 2.5 mg/mL extract has a synergic proliferative effect with dexamethasone, with a cell survival rate evaluated as 155.56%, versus −35.02% for dexamethasone alone. Against staurosporine, the 2.5 mg/mL extract shows cytoprotection, with an increase in cell survival rate, which becomes 53.15% against 36.38% for staurosporine used alone. Furthermore, after an incubation time of 72 h, the 2.5 mg/mL extract shows cytoprotection against dexamethasone, with a strong cytoproliferative effect since the cell survival rate becomes 261.37% versus 30.48% for dexamethasone used alone. Against staurosporine, 2.5 mg/mL of the extract is also effective because it shows cytoprotection and cell proliferation with a cell survival rate of 167.67% instead of 26.67% observed with the only staurosporine. So, after 72 h of incubation, 2.5 mg/mL of extract shows cytoprotective effects against all the substances tested, which includes doxorubicin and staurosporine, both responsible for strong cytotoxicity on bone marrow (Figure 5).

### 3.6. Evaluation of the Efficacy of Acetonitrile–H_2_O Extract in Preventing Doxorubicin-Induced Cytotoxicity on Muscle Fibers

The extract shows no protective effect against doxorubicin, rather it demonstrates a negative synergistic action with this drug, leading to cell death both after 18 h and after 24 h of incubation, with a cell survival rate of 0% (Figure 6a,c). Control cells show cell survival rate of 90% after 18 h of incubation time (Figure 6d).

### 3.7. Evaluation of the Efficacy of Acetonitrile–H_2_O Extract in Preventing Staurosporine-Induced Cytotoxicity on Muscle Fibers

The extract shows no protective effect against staurosporine; on the contrary, it demonstrates a negative synergistic action with this substance, amplifying cell death both after 18 h and after 24 h of incubation, with a cell survival rate reduced to 0%. Again, the extract is of interest for its possible use in enhancing the cytotoxic effect of staurosporine when muscle fibers are the target (Figure 6b,c).

### 3.8. Evaluation of the Efficacy of Acetonitrile–H_2_O Extract in Preventing Doxorubicin-Induced Cytotoxicity on HEK293 Cell Line

The extract shows a protective effect against doxorubicin: after an incubation time of 24 h, the extract at various concentrations had a cytoprotective effect, with a greater effect at higher concentrations in HEK293 cells (Figure 7a). When 5 mg/mL of extract was tested, we had a 98.95% cell survival rate compared to a rate of 56% with doxorubicin alone. Diazoxide has a cytoprotective effect after 24 h of incubation time, with a cell survival rate of 117.75%. After 48 h of incubation, there continues to be a cytoprotective effect, especially where the extract at the highest concentration is tested, with a cell survival rate of 75.48% compared to 14.74% with doxorubicin alone. Cytoprotective effects have maximum evidence after 72 h, where the extract at 5 mg/mL allows a cell survival rate of 119.76% compared to doxorubicin alone, which leads to no cell survival. The extract at lower concentrations shows no cytoprotective effect at 72 h.

### 3.9. Evaluation of the Efficacy of Acetonitrile–H_2_O Extract in Preventing Dexamethasone-Induced Cytotoxicity on the HEK293 Cell Line

After 24 h of incubation, dexamethasone has a proliferative effect, while no cell proliferation is observed in the presence of the extract in HEK293 cells (Figure 7b). After 48 h of incubation, dexamethasone has a strong cytotoxic effect with 0% cell survival, and if the extract is used instead at this incubation time, there is reduced cytotoxicity, with 86% cytoprotection at the highest concentration of the extract (5 mg/mL). Furthermore, after 72 h, dexamethasone no longer shows the cytotoxic effect found at 48 h, suggesting a pathway determining cell death exclusively after 48 h; thus, past this time frame, a normal cell survival condition is reestablished. Moreover, if the extract is used, at 72 h, there is strong cell proliferation, with cell survival more than doubled, with a cell survival rate of 244%. Strong cytotoxic action can be seen if dexamethasone and diazoxide are used in combination, both after 48 h and 72 h.

### 3.10. Evaluation of the Efficacy of Acetonitrile–H_2_O Extract in Preventing Cisplatin-Induced Cytotoxicity on HEK293 Cell Line

The extract shows a cytoprotective effect against cisplatin in HEK293 cells (Figure 7c). After 24 h of incubation, cells treated exclusively with cisplatin have a cell survival rate of 109%; after this time frame, the extract at a concentration of 5 mg/mL shows a further cytoproliferative effect, with a cell survival rate of 130.58%. The effects were concentration-dependent. At 48 h, the 5 mg/mL extract is even more effective, as it allows 86.054% cell survival compared to cells treated exclusively with cisplatin, which showed 0% cell survival, significantly lower than that shown after 24 h. After 72 h of incubation, cisplatin no longer shows a significant cytotoxic effect.

### 3.11. Evaluation of the Efficacy of Acetonitrile–H_2_O Extract in Preventing Irinotecan-Induced Cytotoxicity on the HEK293 Cell Line

The extract shows a protective effect against HEK293 cells treated with irinotecan: the most noticeable effect is with the 5 mg/mL extract, after an incubation time of 24 h, leading to cell proliferation with a cell survival rate of 202.18%, compared with the 80.44% given by the single irinotecan (Figure 7d). The cytoprotective effect is less present after 48 h of incubation, where it is still partly visible with the 5 mg/mL extract, with which there is a cell survival rate of 60.10% at 48 h. After 72 h of incubation, the cytoprotective effect of the extract is no longer present.

### 3.12. Evaluation of the Efficacy of Acetonitrile–H_2_O Extract in Preventing Cisplatin-Induced Cytotoxicity on the SHSY5Y Cell Line

After 24 h of incubation time, cisplatin shows no cytotoxic effect, consequently, we do not evaluate the cytoprotection of the extract in SHSY5Y Cell (Figure 8a). After 48 h of incubation, cisplatin shows a strong cytotoxic effect (0% cell survival rate), and the extract at 2.5 mg/mL concentration shows cytoprotective action. After 72 h of incubation, the only extract at 0.5 mg/mL concentration showed a mild cytoprotective effect. Finally, after 96 h of incubation, the extract at all concentrations tested failed to show a cytoprotective effect.

### 3.13. Evaluation of the Efficacy of Acetonitrile–H_2_O Extract in Preventing Irinotecan-Induced Cytotoxicity on the SHSY5Y Cell Line

After 24 h of incubation, the extracts show a concentration-dependent cytoprotective effect against irinotecan, showing at a 5 mg/mL concentration a cell survival rate of 129%, compared to 60.79% given by irinotecan in SHSY5Y Cell (Figure 8b). Diazoxide shows cytoprotection against irinotecan at this incubation time, with a cell survival rate of 72.07%. At 48 h, as well as after 72 h and 96 h, irinotecan induces massive cytotoxicity, with 0% cell survival. At both 48 h and 72 h, the extract showed cytoprotective effects at the highest concentration of 5 mg/mL. After 96 h of incubation, the extract at all concentrations tested was found to be effective in cytoprotection, particularly the extract at a 0.5 mg/mL concentration, which allows 48.86% cell survival.

### 3.14. Evaluation of the Efficacy of Acetonitrile–H_2_O Extract in Preventing Doxorubicin-Induced Cytotoxicity on the SHSY5Y Cell Line

After 24 h of incubation, the extract at different concentrations is effective in cytoprotection compared with the cytotoxic action of doxorubicin in SHSY5Y Cell (Figure 8c). In particular, the extract at a concentration of 2.5 mg/mL is the one that induces greater cytoprotection, making possible a cell survival rate of 57.92% compared with 37.86% for doxorubicin alone. Diazoxide at this incubation time has a cytoprotective effect, with an increasing survival rate of 50.51%. After 72 h and 96 h of incubation time, doxorubicin has maximum toxicity and zero cell survival. After 72 h and 96 h of incubation, no concentration of the extract is shown to be effective in cytoprotection.

### 3.15. Evaluation of the Efficacy of Acetonitrile–H_2_O Extract in Preventing Dexamethasone-Induced Cytotoxicity on the SHSY5Y Cell Line

Dexamethasone, after 24 h, shows moderate cytotoxicity, and all concentrations of the extract are shown to have a cytoprotective effect in SHSY5Y Cell (Figure 8d). In particular, the most effective one is the extract at 2.5 mg/mL, with a cell survival rate of 95.99%. Furthermore, diazoxide has a cytoprotective action after 24 h of incubation, making the cell survival rate 91.69%. After 72 h of incubation, dexamethasone has no cytotoxic effect.

### 3.16. Liquid Chromatography–Mass Spectrometry (LC-MS) Analysis of the Composition of Extract 2E0217022196DIPFARMTDA

Extract 2E0217022196DIPFARMTDA was analyzed by LC-MS technique, obtaining the spectrometry in Appendix A (Table 1). The molecules identified were obtained using the ChemSpider Advanced Search program. Selecting the protocol used to obtain the spectrometry and the *m*/*z* of the peak, we identified the molecules whose presence is most plausible in our extract. In addition, the article [13] was used as a reference for our hypotheses, since in this work, the phenolic compounds of lentils were analyzed using the LC-MS technique. However, as the preparation of their extract is different and these types of instruments are not universally standardized, only some of our results were comparable. Our analysis focuses on the identification of phenolic compounds, minerals, and sugars; however, Lens culinaris contains other nutrients such as protein in abundance. We do not analyze these compounds in our work because the acetonitrile–water solvent makes them precipitate, not making their composition analyzable. We consider in our analysis only those peaks that show a percentage abundance greater than 2% so that we can hypothesize which substances are present in the greatest quantity in the extract. The *m*/*z* of the peaks is reported with one more unit than shown in the graphs, because in the protocol used, one hydrogen (atomic mass 1.00784) was subtracted, and so we add this missing unit to all peaks. In identifying substances, in some cases, it was necessary to widen the search range to the first decimal place. The peak at 173.091 *m*/*z* has a percent abundance of 3.06%. The substance identified by the peak could be dehydroascorbic acid (oxidized vitamin C), which is strongly present in the human diet and found in many plant foods. The peak observed at 193.0698 *m*/*z* has a percent abundance of 3.82%. Here, the substance we hypothesized, coincident with the peak, is 3-hydroxy-4-methoxycinnamic acid. This substance was also characterized in the referenced article, which further confirms our hypothesis. The peak found at 252.230 *m*/*z* has a percent abundance of 4.78%. Exactly corresponding to this peak, we find three plausible flavonoid structures, specifically flavonic structures. Our hypothesis is again established, flavones being present in the reference work as well. In particular, these molecules have an apigenin-like structure. The peak present at 254.246 *m*/*z* shows a percent abundance of 3.22%. Precisely, this peak corresponds to a flavonoid-like structure, specifically a flavanone, similar to naringenin. Here again, our hypothesis is concordant with what is described in the referenced article. The peak found at 278.246 *m*/*z* shows a percent abundance of 6.06%, and here, we recognize three plausible flavonoid-type structures, particularly flavones. The peak observed at 292.225 *m*/*z* has a percent abundance of 5.21%, which identifies one particular catechin: catechin-2,3,4-^13^C_3_. The reference article confirms our hypothesis, reporting the same compound. The peak identified at 341.1070 *m*/*z* has a percent abundance of 18.07%, and here we recognize the structure of mellitic acid, which we assume to be the molecule described by this peak. The peak present at 355.122 *m*/*z* has a percent abundance of 2.72%; corresponding to the peak, we identify chebulic acid, described as an important antioxidant at the cellular level. The peak found at 377.0849 *m*/*z* has a percent abundance of 4.26%, and associated with this peak is trehalose, a sugar found in several types of plants and foods, which possesses the ability to protect cell membranes and proteins from damage and denaturation. The peak found at 417.102 *m*/*z* has a percent abundance of 2.38%; corresponding to this peak, we observe kaempferol 3-O-arabinoside, an antioxidant flavonoid. Kaempferol was also detected in the reference article but linked to a different sugar. The peak observed at 517.175 *m*/*z* has a percent abundance of 4.05%; in this case, we recognize apigenin 7-O-(6-malonyl-β-D-glucoside). Apigenin is present in the reference article, although differently substituted here. This substance is always of the flavonoid type; precisely, it is a flavone (Table 1).

Since we do not use the primary standards of the substances we refer to in our analysis, what we have found in our study needs to be possibly confirmed later on.

### 3.17. Liquid Chromatography–Mass Spectrometry (LC-MS) Analysis of the Composition of Extract 1E030521DIPFARMTDA

The hydroalcoholic extract 1E030521DIPFARMTDA, which was investigated in the previous study, was analyzed by the LC-MS technique, obtaining the spectrometry in Appendix A . We used ChemSpider Advanced Search to identify the molecules related to the peaks (Table 2). We made a comparison between the molecules obtained from the LC-MS of the acetonitrile–water extract and the molecules obtained from this hydroalcoholic extract, so we can understand the different cell survival results related to the two different extracts. Since the protocol used in the execution of the LC-MS was the same, the *m*/*z* of the peaks is reported with one more unit than shown in the graphs. The peak obtained at 146.048 *m*/*z* has a percent abundance of 6.57%, and this is identified as butyric acid acetate; butyric acid esters are used as food additives. The peak at 173.095 *m*/*z* has a percent abundance of 8.92%, and the substance identified by the peak matches dehydroascorbic acid (oxidized vitamin C), which was also found in the acetonitrile–water extract. The peak observed at 179.058 *m*/*z* has a percent abundance of 5.08%, and the substance coincident with the peak is D-glucopyranose, which is the most popular sugar found in nature, like fructose. The peak found at 195.053 m/z has a percent abundance of 6.45%, and the molecule which matches here is D-gluconic acid, a substance that occurs often in natural products and is also used as a food additive. The peak present at 207.09 *m*/*z* shows a percent abundance of 4.25%; this peak corresponds to hydroxy citric acid, a derivative of citric acid that is found in a variety of plants. The peak found at 255.235 m/z shows a percent abundance of 22.91%, and this peak corresponds to phenyl-D-galactopyranoside, which is a substituted galactoside. The peak observed at 277.220 *m*/*z* has a percent abundance of 18.3%, identified as 6-O-(4-Hydroxy-2-methylenebutanoyl)-β-D-glucopyranose, which is a substituted molecule of glucopyranose. The peak found at 278.225 *m*/*z* shows a percent abundance of 3.57%, and here we recognize the same flavonoid-type structures found in LC-MS of ACN-H2O extract, particularly flavones. The peak present at 295.231 *m*/*z* has a percent abundance of 4.76%; corresponding to the peak is medicagol, which belongs to the class of organic compounds known as coumestans, so this is also a flavonoid that is found in several different foods. The peak found at 341.112 *m*/*z* has a percent abundance of 10.8%, and here we recognize the structure of mellitic acid, which was also found in the precedent analysis. The peak found at 377.089 *m*/*z* has a percent abundance of 24.84%; associated with this peak is trehalose, a sugar owner of cell protection properties found also in the precedent analysis. The peak observed at 379.087 *m*/*z* has a percent abundance of 8.35%; in this case, we recognize pelentanic acid, which is a coumarinyl acid. The peak found at 417.108 *m*/*z* has a percent abundance of 11.82%; corresponding to this peak, we observe kaempferol 3-O-arabinoside, an antioxidant flavonoid that was also found in the precedent analysis. The peak observed at 517.182 *m*/*z*, which has a percent abundance of 6.57%, is a peak found in the previous analysis that corresponds to apigenin 7-O-(6-malonyl-β-D-glucoside). The peak observed at 701.516 *m*/*z* has a percent abundance of 36.67% and it is associated with galactopyranosyl-D-glucose hydrate, a sugar molecule. The peak found at 723.501 *m*/*z*, with a percent abundance of 83.97%, corresponds to a molecule of glucopyranose with many substitutions, which is 6-O-β-D-glucopyranosyl-1-O-[(2E)-3-(4-hydroxy-3,5-dimethoxyphenyl)-2-propenoyl]-2-O-[(2E)-3-(4-hydroxy-3-methoxyphenyl)-2-propenoyl]-β-D-glucopyranose. Finally, the peak found at 725.516 *m*/*z* with a percent abundance of 100% is another molecule of glucopyranose with substitutions, which is 5,7-dihydroxy-2-(4-hydroxyphenyl)-4-oxo-4H-chromen-3-yl 6-deoxy-α-L-mannopyranosyl-(1->3)-[β-D-xylopyranosyl-(1->2)]-α-D-glucopyranoside.

In this LC-MS analysis, the main difference we have found is a better extraction capacity of sugars, obviously due to the nature of the hydroalcoholic solvent.

## 4. Discussion

It was demonstrated in this work that acetonitrile–water extract (concentration range 0.1–5 mg/mL) generally prevented cytotoxic damage induced by cisplatin, irinotecan, staurosporine, and doxorubicin on murine bone marrow cells and osteoblasts, and also HEK293 and SHSY5Y cell lines. In fact, we found that on skeletal muscle fibers, the extract increased the cytotoxicity of chemotherapeutics. More specifically, the following main effects were observed.

On osteoblasts, the 2.5 mg/mL extract was effective as cytoprotective against doxorubicin after an incubation time of 24 h, 48 h, and especially after 72 h, where the cell survival rate was brought to 53.601% by the extract, and 0% using doxorubicin alone. The 2.5 mg/mL extract was also effective against irinotecan, with a cell survival rate of 97.365%, versus 86.523% using irinotecan alone after 72 h of incubation. Finally, the 2.5 mg/mL extract has cytoprotective effects against the unselective multi-tyrosine kinase inhibitor staurosporine, after incubation times of 48 and 72 h, showing the best results at the longest incubation, with a cell survival rate of 89.96% versus 10.25% using staurosporine alone.

Bone marrow cells have different behavior depending on incubation times and the use of different chemotherapeutics. The 2.5 mg/mL extract shows cytoprotection after 72 h of incubation, increasing cell survival rate to 145.402% from −86.078% for doxorubicin used alone, but at a longer incubation time, the 2.5 mg/mL extract increases the cytotoxicity of doxorubicin, obtaining a cell survival rate of −61.684% versus −10.56% achieved with doxorubicin alone. However, it has cytoprotective effect on bone marrow cells against dexamethasone and staurosporine; for instance, against staurosporine, the extract shows a cytoprotective effect, bringing the cell survival rate to 167.67% from 26.67% obtained with staurosporine alone. Thus, the most important result we achieved is that the 2.5 mg/mL extract has cytoprotective properties if used after 72 h of incubation, against all the substances tested.

On native Flexor digitorum brevis muscle fibers, both doxorubicin and staurosporine have a cytotoxic effect after incubation times of 18 h and 24 h. Unexpectedly, if the extract is also administered with these two substances, greater cytotoxicity is observed, with a 0% cell survival rate at both 18 h and 24 h. These results may suggest a future investigation on tumor-like muscle fibers, where the extract could enhance cytotoxicity if its activity is demonstrated in tumor cells.

On renal Hek293 and SHSY5Y neuroblastoma cells, the extract was tested and co-administered with the substances cisplatin, irinotecan, doxorubicin, and dexamethasone. It was effective against these drugs in these cells lines at different incubation times at high concentrations (2.5–5 mg/mL), with some exceptions—indeed, doxorubicin has strong cytotoxicity in the SHSY5Y neuroblastoma cells, and the extract shows a cytoprotective effect only after an incubation time of 24 h at a 2.5 mg/mL concentration.

Thus, on the last two cell types, the development of the present extract could be implemented in the research area related to the chemoprevention of renal and neuronal damage by chemotherapeutics such as cisplatin, irinotecan, and doxorubicin.

Liquid chromatography–mass spectrometry (LC-MS) analysis on the acetonitrile–water extract showed the presence of many phenolic structures, especially flavonoids of various types such as flavones, flavonols, catechins, kaempferol, apigenin, and other substances such as 3-hydroxy-4-methoxycinnamic acid, trehalose, dehydroascorbic acid, mellitic acid, and chebulic acid.

Flavonoids are the compounds with the highest percentage abundance in our extract. The structure of flavonoids is based on the flavonoid core, which consists of three phenolic rings referred to as A, B, and C rings. The benzene ring A is condensed with a six-membered ring (C), which in position two carries a phenylbenzene ring (B) as a substituent. The C ring can be a heterocyclic pyran, which produces flavanols (catechins) and anthocyanidins, or pyrone, which produces flavonols, flavones, and flavanones. The term 4-oxo-flavonoids is often used to describe flavonoids, such as flavanols (catechins), flavanones, flavonols, and flavones, which carry a carbonyl group on C-4 of the C ring [14] (Appendix A).

Abundant in our extract are flavones and flavanones. Flavones are detected at the peaks at 253.230 *m*/*z* and 279.246 *m*/*z*. In particular, two of the flavones observed at peak 253.230 *m*/*z* possess a structure similar to apigenin (Appendix A). In addition, corresponding to peaks at 517.175 *m*/*z*, we observe apigenin 7-O (6-malonyl-β-D-glucoside). Among the wide variety of phenolic compounds, apigenin is one of the most renowned ones, with countless nutritional and organoleptic characteristics. In addition, it possesses beneficial health properties, which could lead to its possible inclusion in nutraceutical formulations. So far, few trials report adverse metabolic reactions of apigenin; therefore, its consumption through the diet is recommended. After ingestion, it goes through several metabolic pathways to exert its healing properties, and its pharmacokinetic behavior influences its tissue distribution and bioactivity. In nature, apigenin is also found bound through C-C or C-O-C bonds of dimeric forms. There are different pharmacokinetic behaviors between flavonoid aglycones and their glycosides. The effect of O-glycosylation or C-glycosylation of apigenin can affect its metabolism in different ways and thus influence its antioxidant potential and benefits in neurodegeneration and cancer [15] (Appendix A).

At a peak of 255.246 *m*/*z* is a flavanone structure similar to naringenin (Appendix A), a flavonoid belonging to the subclass of flavanones. Naringenin is endowed with broad biological effects for human health. It has also been reported to have the ability to modulate signaling pathways related to fatty acid metabolism and to promote fatty acid oxidation, decreasing lipid accumulation in the liver and thus preventing hepatic steatosis, as well as decreasing plasma lipid and lipoprotein accumulation. In addition, anti-cancer effects have also been attributed to this metabolite, mainly related to its ability to repair DNA. Exposure of cells to naringenin for 24 h resulted in a 24% reduction in DNA hydroxylation damage. In addition, antiviral effects have been reported; for example, it inhibits the assembly and long-term production of infectious hepatitis C virus particles [16].

Another abundant molecule in the extract turned out to be mellitic acid, detected at the peak at 342.1070 *m*/*z*; this is a carboxylic acid present in the mellite mineral. With Lens culinaris being mineral-rich, it is plausible to find this substance in the extract. PubChem reports this substance as an irritant, as it has three GHS hazard statements: H315, H319, and H335. Nevertheless, similar to bisphosphonates, mellitic acid has a strong affinity for calcium phosphate minerals and can inhibit apatite crystallization both in vitro and in vivo. In animal experiments, matrix-induced ectopic osteogenesis and cartilage formation developed with the application of mellitic acid. High local concentrations of mellitic acid, which might be more effective in the therapy of ectopic ossifications, are problematic because of the possible formation of crystalline deposits in the tissue [17] (Appendix A).

Catechin-2,3,4-^13^C_3_, corresponding to a peak of 293.225 *m*/*z*, is also a substance highly present in the extract, observed in the reference article. Catechins are polyphenolic compounds, especially present in Camellia sinensis green tea extract, and of these, epigallocatechin-3-gallate (EGCG) accounts for about 59% of the total catechins present in the medicinal plant [18] (Appendix A). EGCG ((-)—epigallocatechin-3-gallate) has a polyphenolic structure with tri-hydroxyl substitution on the B ring and a gallate moiety esterified to carbon 3 on the C ring [19]. These structural features contribute to its pronounced activity as an antioxidant and iron-chelating agent, and it also has several pharmacological and biological properties: antioxidant, free radical scavenger [20], antibacterial [21], antiviral [22], anti-diabetic [23], cardioprotective, anti-atherosclerotic, anti-inflammatory [24], neuroprotective [25], and anti-carcinogenic [26]. It has been intensively studied in both cell culture and animal experiments, and epidemiological and clinical studies [26]. However, limitations in terms of stability and bioavailability have hindered its application in clinical settings. Several studies have already reported the use of nanoparticles as EGCG delivery vehicles for cancer therapy [19].

Trehalose (Appendix A) is another substance present in abundance in our sample, corresponding to a peak of 378.0849 *m*/*z*. This compound was originally identified as an agent that induces cellular macro autophagy, which is the breakdown and recycling of old or damaged macromolecules that occurs in response to cellular stress. This mechanical action was initially suspected because of the protective effects of trehalose against β-amyloid aggregation in neurological cell lines. Because autophagy regulates cellular and whole-body metabolic homeostasis, trehalose is believed to attenuate metabolic diseases by activating autophagy in specific tissue compartments. Recently, inhibition of cellular stress pathways, such as matrix metal proteinases and the p38 MAPK pathway, has been observed. Several animal models have shown that trehalose protects against atherosclerosis and dyslipidemia. Overall, recent data examining the effects of trehalose in murine metabolic diseases demonstrate extensive effects of trehalose when taken orally and parenterally in atherosclerotic disease, cardiac remodeling, dyslipidemia, and hepatic steatosis. Because trehalose is “generally recognized as safe” by the Food and Drug Administration, recent successes in treating metabolic diseases in animals have led to its use in humans. Overall, acute exposure to this substance in humans reduced glycemic and insulinotropic responses compared with sucrose. Human exposure modestly improves cardiometabolic risk factors and vascular function in selected populations [27].

Another substance in the extract is 3-hydroxy-4-methoxycinnamic acid (isoferulic acid), observed at 194.069 *m*/*z*, also found in the reference article (Appendix A). Cinnamic acid, a naturally occurring aromatic carboxylic acid, is a basic chemical found in plants such as Cinnamomum cassia (Chinese cinnamon) and Panax ginseng, and more generally in fruits, whole grains, vegetables, and honey. The presence of a substituted acrylic acid group on the phenyl ring gives cinnamic acid a cis or trans configuration; the trans configuration is the more common of the two. Studies have reported that cinnamic acid has antioxidant, antimicrobial, antitumor, neuroprotective, anti-inflammatory, and antidiabetic properties [28]. Cinnamic acid terminates radical chain reactions by donating electrons that react with radicals to form stable products. Cinnamic acid can be prepared by enzymatic deamination of phenylalanine. In addition to cinnamic acid derivatives occurring naturally in plants, the presence of a benzene ring and acrylic acid group allows it to be modified, resulting in synthetic derivatives of cinnamic acid. In addition to being intermediates to produce other compounds such as stilbenes and styrene, cinnamic acid derivatives exhibit antibacterial, antifungal, anti-inflammatory, neuroprotective, anticancer, and antidiabetic activities. The cinnamic acid derivatives reported by researchers include ferulic acid, curcumin, caffeic acid, p-hydroxycinnamic acid, and coumaric acid [28]. The acid we identified is a substituted cinnamic acid, specifically isoferulic acid. The inhibitory activity of ferulic acid on fructose- and glucose-mediated protein glycation and oxidation of bovine serum albumin was proposed. Therefore, isoferulic acid could be a promising new anti-glycation agent for the prevention of diabetic complications through inhibiting the formation of advanced glycation end products (AGEs) and oxidation-dependent protein damage [29].

From our investigation of the extract, dehydroascorbic acid appears to be present, identified by the peak at 174.091 *m*/*z*. Dehydroascorbic acid is the oxidized, reversible form of ascorbic acid (vitamin C). Although ascorbic acid is the more reduced species of the two, dehydroascorbic acid appears to have antioxidant properties of its own, in addition to those of ascorbic acid (Appendix A). For example, dehydroascorbic acid has been shown to have greater activity than ascorbic acid in protecting low-density lipoproteins from oxidation by copper ions [30]. Dehydroascorbic acid reacts with homocysteine thiolactone, converting it into the toxic compound 3-mercaptopropionaldehyde [31]. Taken together, these results suggest that rapidly dividing tumor cells produce unusually high amounts of homocysteine thiolactone and that the administered dehydroascorbic acid infiltrates the cells and converts thiolactone to mercaptopropionaldehyde, which kills tumor cells. The indicated mechanism has obvious advantages for cancer therapy; the toxic agent is generated within cancer cells and appears to be selective for cancer cells [31].

Another compound found within the extract is chebulic acid (Appendix A), corresponding to the peak present at 356.122 *m*/*z*. The antioxidant activities of chebulic acid were evaluated by two different methods and compared with those of epigallocatechin gallate (EGCG). The results of one of the two assays showed that the antioxidant activity of chebulic acid was 2.5 times lower than that of EGCG, but in the second assay, the activities were similar. In addition, the antioxidant activity of chebulic acid was further evaluated by the ESR spin-trapping technique to examine its superoxide radical scavenger activity. It was observed that oxidative stress causes damage to cellular and extracellular macromolecules, such as proteins, lipids, and nucleic acids, resulting in a shift in the balance toward the pro-oxidant state. Therefore, antioxidants are closely related to the prevention of degenerative diseases, including cardiovascular and neurological diseases [32].

Kaempferol 3-O-arabinoside (Appendix A) is the molecule with the lowest percentage abundance considered, corresponding to the peak at 418.102 m/z. This is an antioxidant flavonoid, yet on the PubChem portal, hazard statements H301 (toxic if inhaled) and H341 (suspected of generating genetic defects) are attributed to it; however, it is inactive in almost all biological tests related to toxicity, except those related to NSD2 (SET domain protein binding to nuclear receptor 2) and EZH2 (subunit of the polycomb repressive complex EZH2 (enhancer of zeste 2)). However, kaempferol has many beneficial effects due to its antioxidant capacity. It is a tetrahydroxyflavone found in various parts of plants, such as seeds, leaves, fruits, flowers, and vegetables [33]. Kaempferol and its glycosylated derivatives are cardioprotective, neuroprotective, anti-inflammatory, antidiabetic, antioxidant, antimicrobial, and anticancer agents [34]. Epidemiological studies have shown that high kaempferol intake is associated with a reduced incidence of several types of cancer, including cancer in organs such as the skin, liver, colon, ovary, pancreas, stomach, and bladder [35]. Kaempferol inhibits various tumor lines by triggering apoptosis through different mechanisms, such as cell cycle arrest in the G2/M phase, downregulation of signaling pathways, activation of caspase−3, −7, and −9, or inhibition of angiogenesis. Kaempferol and its glycosides, as well as plants containing this substance, have also demonstrated potent antioxidant activity both in vivo and in vitro, capable of reducing ROS concentrations [36]. At submicromolar concentrations, it also inhibits pro-oxidant enzymes such as xanthine oxidase [37] and activates antioxidant enzymes such as superoxide dismutase, catalase, and heme oxygenase-1, even preventing the generation of hydroxyl radicals by chelating copper and iron ions [38]. However, data on the long-term effect of kaempferol intake are still insufficient, and although the low bioavailability of kaempferol is a significant obstacle, the use of kaempferol-based nanoparticles has brought more hope for cancer prevention chemo strategies [39].

Some of the molecules observed were also present in the composition of the hydroalcoholic extract, according to LC-MS analysis. We found in both the extracts dehydroascorbic acid, flavonoids (in particular, flavons), mellitic acid, trehalose, kaempferol 3-O-arabinoside, and apigenin 7-O-(6-malonyl—β-D-glucoside). The cytoprotective effects shown by both extracts on SHSY5Y and HEK293 could be attributed to these molecules. Moreover, molecules that were only found in acetonitrile–water extract could be responsible for the cytoprotection of osteoblasts and bone marrow cells, since the hydroalcoholic extract does not have protective effects on these cellular types, except for dexamethasone on osteoblasts. Those molecules are 3-hydroxy-4-methoxycinnamic acid, chebulic acid, and catechin-2,3,4-^13^C_3_. Furthermore, the major cytotoxicity on muscle fibers caused by acetonitrile–water extract could be triggered by a higher concentration of mellitic acid (18.07%) found, associated with lower concentrations of dehydroascorbic acid (3.06%), trehalose (4.26%), kaempferol 3-O-arabinoside (2.38%), and apigenin 7-O-(6-malonyl-β-D-glucoside) (4.05%) in comparison to the hydroalcoholic extract.

In addition, we tested the effect of diazoxide, which is a well-known potassium channel opener specifically acting on ATP-sensitive K+ channels with cytoprotective actions in neurons, pancreatic beta cells, skeletal muscle, and in cell lines expressing the recombinant subunits [40,41,42,43,44,45,46,47,48]. Here, diazoxide was effective in preventing the cytotoxicity induced by dexamethasone, irinotecan, cisplatin, and staurosporine in the cell lines and murine osteoblasts against staurosporine, but it was less effective than the novel extract against doxorubicin in bone marrow cells.

## 5. Conclusions

Acetonitrile–water extract could be used for cytoprotection of osteoblasts, bone marrow cells, and neuronal and renal cellular lines. Moreover, it could be used as an enhancer of chemotherapeutics on sarcomas, due to its cytotoxic action on muscle fibers, but maintaining a cytoprotective action on osteoblasts and bone marrow cells. This could be assumed if the same results we obtained on healthy muscle fibers are obtained on tumoral muscle fibers.

A limitation of our work is the fact that we did not quantify the relative molecules present in the extract, but this is because the observed effects can be related to the phytocomplexes’ composition rather than to a specific molecule.

In addition, we did not investigate the signaling pathways responsible for the cytoprotection caused by the extract, since multiple pathways are expected to be involved in these actions.

## 6. Patents

No patent is resulting from the work reported in this manuscript.

## Figures and Tables

**Figure 1 cells-12-00575-f001:**
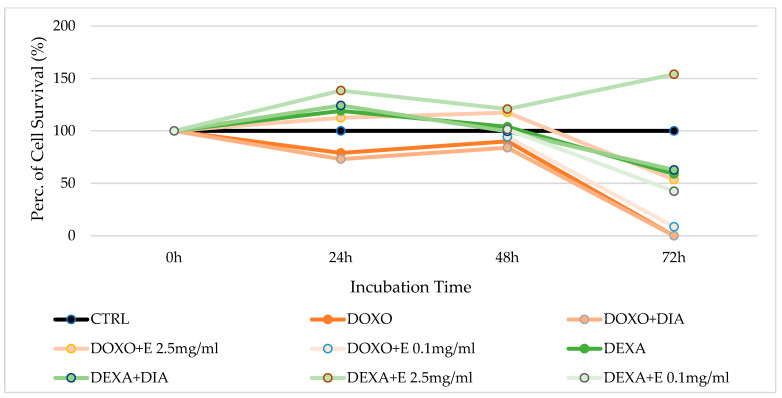
Percentage of cell growth values of osteoblasts co-incubated with two different drugs such as doxorubicin (DOXO) and dexamethasone (DEXA), each drug and diazoxide (DIA), and each drug and various concentrations of the extract (E) measured using the crystal violet assay, compared to controls (CTRL). Data are shown for two different concentrations of the extract (2.5 mg/mL and 0.1 mg/mL) after 0 h, 24 h, 48 h, and 72 h of incubation, using the drug doxorubicin in the orange lines and dexamethasone in the green lines. One-way ANOVA testing for differences between and within groups of treatments showed significant differences between treatments (F > 2 and *p* < 0.005) at different incubation times for DOXO and DOXO + E 2.5, and DEXA after 72 h and DEXA + E 2.5 (N experiments = 6).

**Figure 2 cells-12-00575-f002:**
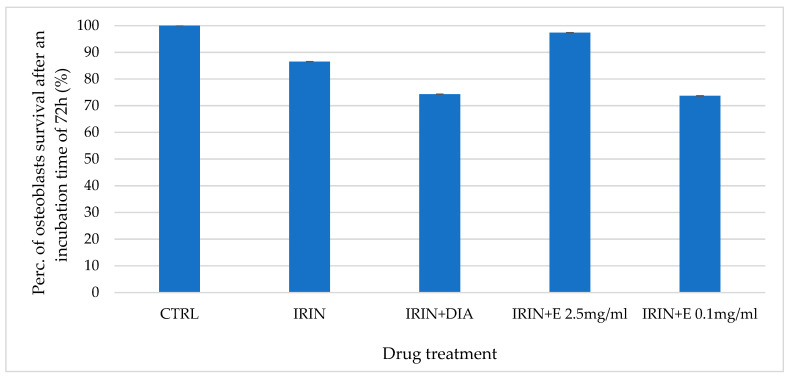
Percentage of cell growth values of osteoblasts co-incubated with irinotecan (IRIN) alone, irinotecan and diazoxide (DIA), and irinotecan and two different concentrations of the extract (E) measured using the crystal violet assay, compared to control (CTRL). Data are shown for two different concentrations of the extract (2.5 mg/mL and 0.1 mg/mL) after 72 h of incubation. One-way ANOVA testing for differences between and within groups of treatments showed significant differences between treatments (F > 1.8 and *p* < 0.05) for IRIN and IRIN + E 2.5 (N experiments = 6).

**Figure 3 cells-12-00575-f003:**
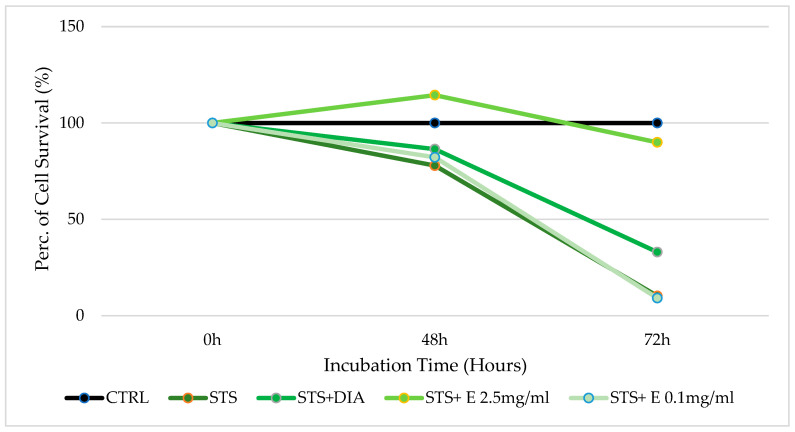
Percentage of cell growth values of osteoblasts co-incubated with staurosporine (STS) alone, staurosporine and diazoxide (DIA), and staurosporine and two different concentrations of the extract (E) measured using the crystal violet assay, compared to controls (CTRL). Data are shown for two different concentrations of the extract (2.5 mg/mL and 0.1 mg/mL) after 0 h, 24 h, 48 h, and 72 h of incubation. One-way ANOVA testing for differences between and within groups of treatments showed significant differences between treatments (F > 1.9 and *p* < 0.005) at different incubation times for STS and STS + E 2.5, and DIAZO after 72 h (N experiments = 6).

**Figure 4 cells-12-00575-f004:**
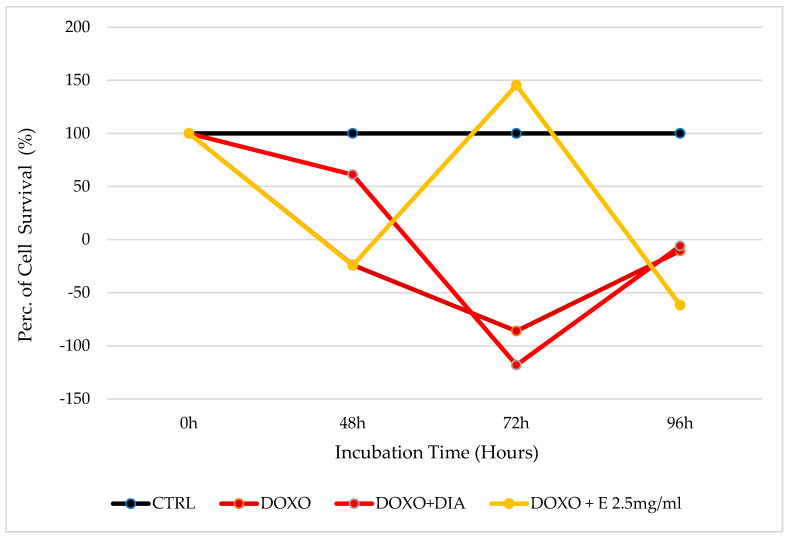
Percentage of cell growth values of bone marrow cells co-incubated with doxorubicin (DOXO) alone, doxorubicin and diazoxide (DIA), and doxorubicin and the extract (E) measured using the crystal violet assay, compared to control (CTRL). Data are shown for a 2.5 mg/mL concentration of the extract after 0 h, 48 h, 72 h, and 96 h of incubation. One-way ANOVA testing for differences between and within groups of treatments showed significant differences between treatments (F > 2.5 and *p* < 0.005) at different incubation times for DOXO and DOXO + E 2.5 after 72 h of incubation (N experiments = 6). The negative values in the figure were due to the calculation made relative to the percentage values of the number of cells in each well, which in turn were derived through the calibration curve obtained from a growth curve in the 96-well plates. Thus, when a negative number for the percentage of cell survival is obtained, this means that there is a very low absolute value.

**Figure 5 cells-12-00575-f005:**
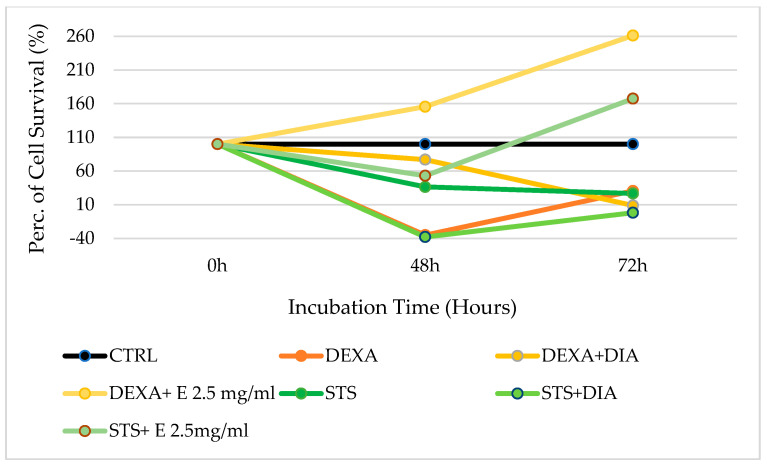
Percentage of cell growth values of bone marrow cells co-incubated with dexamethasone (DEXA) and staurosporine (STS), in the presence of diazoxide (DIA), and the extract (E) with a concentration of 2.5 mg/mL measured using the crystal violet assay, compared to control (CTRL). Data are shown for a 2.5 mg/mL concentration of the extract after 48 h and 72 h of incubation. One-way ANOVA testing for differences between and within groups of treatments showed significant differences between treatments (F > 1.9 and *p* < 0.005) for DEXA and DEXA 2.5 E , STS and STS + E 2.5 after 48 and 72 h (N experiments = 6). The negative values in the figure were due to the calculation made relative to the percentage values of the number of cells in each well which in turn were derived through the calibration curve obtained from a growth curve in the 96-well plates. Thus, when a negative number for the percentage of cell survival is obtained, this means that there is a very low absolute value.

**Figure 6 cells-12-00575-f006:**
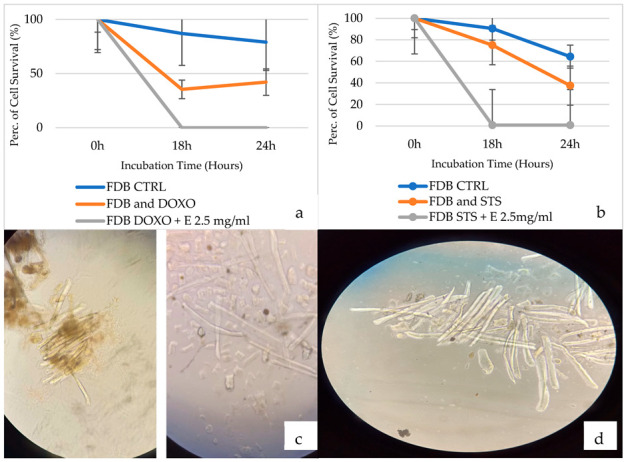
(**a**) Cell survival rate at different incubation times (18 h and 24 h) of mouse Flexor digitoturum brevis (FDB) muscle fibers treated with doxorubicin (DOXO) and with doxorubicin associated with the acetonitrile–water extract E (2.5 mg/mL) (N experiments = 3). (**b**) Cell survival rate at different incubation times (18 h and 24 h) of mouse FDB treated with staurosporine (STS) and with staurosporine associated with acetonitrile–water extract E (2.5 mg/mL) (N = 3 experiments). (**c**) On the left, an evaluation of FDB fiber survival after an incubation time of 18 h with staurosporine (STS) (2.14 × 10^−6^ M). On the right is an evaluation of FDB fiber survival after an incubation time of 18 h with doxorubicin (1 × 10^−6^ M). The fiber survival was evaluated by direct visual inspection and cell counting under Zeiss Axiovert 10 inverted microscope (×10). For each fiber, three individual measurements were performed at three or two different points. The appearance of multiple sarcolemma blebs often preceded cellular death. (**d**) Sample of untreated muscle fibers used as a control after 18 h of incubation time.

**Figure 7 cells-12-00575-f007:**
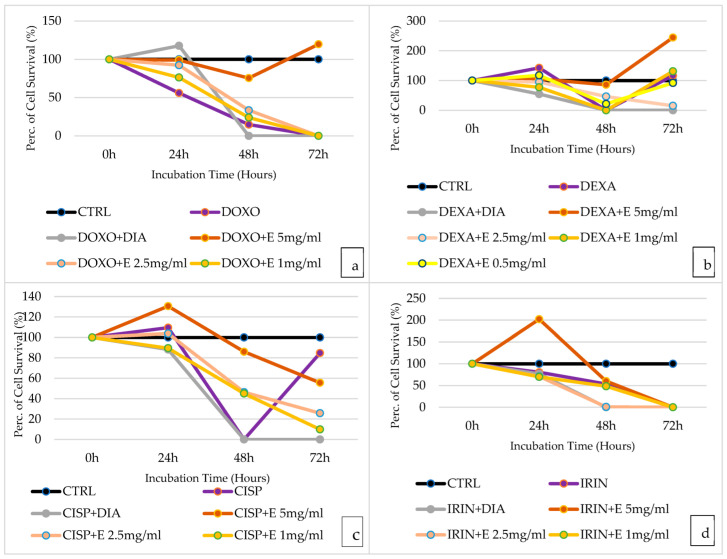
(**a**) Percentage of cell growth values of HEK293 cells co-incubated with doxorubicin (DOXO), doxorubicin and diazoxide (DOXO + DIA), and doxorubicin and various concentrations of the extract (DOXO + E) measured using the crystal violet assay, compared to controls (CTRL). Data are shown for various concentrations of the extract (5 mg/mL, 2.5 mg/mL, 1 mg/mL) after 0 h, 24 h, 48 h, and 72 h of incubation, using the drug doxorubicin (DOXO) in the purple lines, doxorubicin and diazoxide (DOXO + DIA) in gray lines, and doxorubicin with the various concentrations of the extract (DOXO + E) in orange lines. (**b**) Percentage of cell growth values of HEK293 cells co-incubated with dexamethasone (DEXA), dexamethasone and diazoxide (DEXA + DIA), and dexamethasone and various concentrations of the extract (DEXA + E) measured using the crystal violet assay, compared to controls (CTRL). Data are shown for various concentrations of the extract (5 mg/mL, 2.5 mg/mL, 1 mg/mL, 0.5 mg/mL) after 0 h, 24 h, 48 h, and 72 h of incubation, using the drug dexamethasone (DEXA) in the purple lines, dexamethasone and diazoxide (DEXA + DIA) in gray lines, and dexamethasone with the various concentrations of the extract (DEXA + E) in orange and yellow lines. (**c**) Percentage of cell growth values of HEK293 cells co-incubated with cisplatin (CISP), cisplatin and diazoxide (CISP + DIA), and cisplatin and various concentrations of the extract (CISP + E) measured using the crystal violet assay, compared to controls (CTRL). Data are shown for various concentrations of the extract (5 mg/mL, 2.5 mg/mL, and 1 mg/mL) after 0 h, 24 h, 48 h, and 72 h of incubation, using the drug cisplatin (CISP) in the purple lines, cisplatin and diazoxide (CISP + DIA) in gray lines, and cisplatin with the various concentrations of the extract (CISP + E) in orange lines. (**d**) Percentage of cell growth values of HEK293 cells co-incubated with irinotecan (IRIN), irinotecan and diazoxide (IRIN + DIA), and irinotecan and various concentrations of the extract (IRIN + E) measured using the crystal violet assay, compared to controls (CTRL). Data are shown for various concentrations of the extract (5 mg/mL, 2.5 mg/mL, and 1 mg/mL) after 0 h, 24 h, 48 h, and 72 h of incubation, using the drug irinotecan (IRIN) in the purple lines, irinotecan and diazoxide (IRIN + DIA) in gray lines, and irinotecan with the various concentrations of the extract (IRIN + E) in orange lines.

**Figure 8 cells-12-00575-f008:**
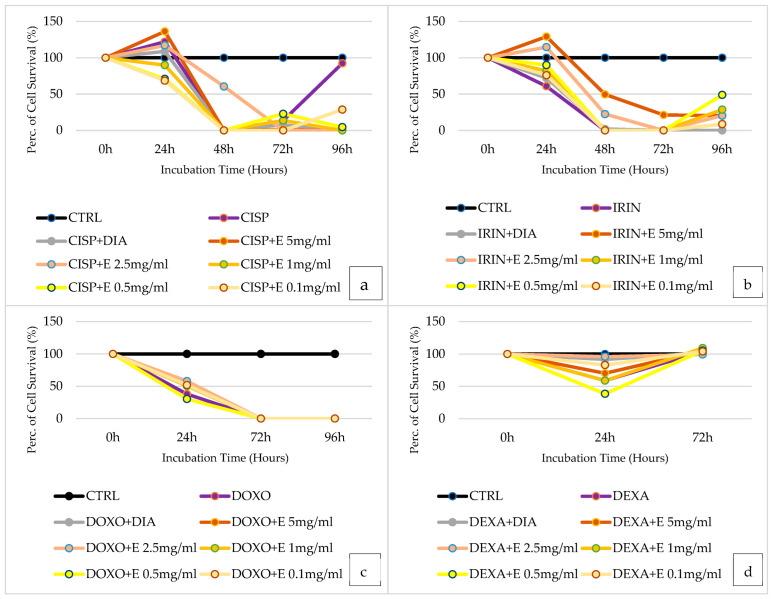
(**a**) Percentage of cell growth values of SHSY5Y cells co-incubated with cisplatin (CISP), cisplatin and diazoxide (CISP + DIA), and cisplatin and various concentrations of the extract (CISP + E) measured using the crystal violet assay, compared to controls (CTRL). Data are shown for various concentrations of the extract (5 mg/mL, 2.5 mg/mL, 1 mg/mL, 0.5 mg/mL, and 0.1 mg/mL) after 0 h, 24 h, 48 h, 72 h, and 96 h of incubation, using the drug cisplatin (CISP) in the purple lines, cisplatin and diazoxide (CISP + DIA) in gray lines, and cisplatin with the various concentrations of the extract (CISP + E) in orange and yellow lines. (**b**) Percentage of cell growth values of SHSY5Y cells co-incubated with irinotecan (IRIN), irinotecan and diazoxide (IRIN + DIA), and irinotecan and various concentrations of the extract (IRIN + E) measured using the crystal violet assay, compared to controls (CTRL). Data are shown for various concentrations of the extract (5 mg/mL, 2.5 mg/mL, 1 mg/mL, 0.5 mg/mL, and 0.1 mg/mL) after 0 h, 24 h, 48 h, 72 h, and 96 h of incubation, using the drug irinotecan (IRIN) in the purple lines, irinotecan and diazoxide (IRIN + DIA) in gray lines, and irinotecan with the various concentrations of the extract (IRIN + E) in orange and yellow lines. (**c**) Percentage of cell growth values of SHSY5Y cells co-incubated with doxorubicin (DOXO), doxorubicin and diazoxide (DOXO + DIA), and doxorubicin and various concentrations of the extract (DOXO + E) measured using the crystal violet assay, compared to controls (CTRL). Data are shown for various concentrations of the extract (5 mg/mL, 2.5 mg/mL, 1 mg/mL, 0.5 mg/mL, and 0.1 mg/mL) after 0 h, 24 h, 48 h, 72 h, and 96 h of incubation, using the drug doxorubicin (DOXO) in the purple lines, doxorubicin and diazoxide (DOXO + DIA) in gray lines, and doxorubicin with the various concentrations of the extract (DOXO + E) in orange and yellow lines. (**d**) Percentage of cell growth values of SHSY5Y cells co-incubated with dexamethasone (DEXA), dexamethasone and diazoxide (DEXA + DIA), and dexamethasone and various concentrations of the extract (DEXA + E) measured using the crystal violet assay, compared to controls (CTRL). Data are shown for various concentrations of the extract (5 mg/mL, 2.5 mg/mL, 1 mg/mL, 0.5 mg/mL, and 0.1 mg/mL) after 0 h, 24 h, and 72 h of incubation, using the drug dexamethasone (DEXA) in the purple lines, dexamethasone and diazoxide (DEXA + DIA) in gray lines, and dexamethasone with the various concentrations of the extract (DEXA + E) in orange and yellow lines.

**Table 1 cells-12-00575-t001:** Table showing substances obtained by the most important peaks with LC-MS of the acetonitrile–water extract, with relative abundance percentage and molecular ion (*m*/*z*) + 1.00784 (atomic mass of hydrogen).

Compound	Abundance %	Molecular Ion (*m*/*z*) + 1.00784 (Hydrogen Atomic Mass)
Dehydroascorbic Acid	3.06%	174.091
3-Hydroxy-4-methoxycinnamic acid	3.82%	194.069
Flavons	4.78%	253.230
Flavanones	3.22%	255.246
Flavones	6.06%	279.246
Catechin-2,3,4-^13^C_3_	5.21%	293.225
Mellitic acid	18.07%	342.1070
Chebulic acid	2.72%	356.122
Trehalose	4.26%	378.0849
Kaempferol 3-O-arabinoside	2.38%	418.102
Apigenin 7-O-(6-malonyl-β-D-glucoside)	4.05%	518.175

**Table 2 cells-12-00575-t002:** Table showing substances obtained by the most important peaks with LC-MS of the hydroalcoholic extract, with relative abundance percentage and molecular ion (*m*/*z*) + 1.00784 (atomic mass of hydrogen).

Compound	Abundance %	Molecular Ion (*m*/*z*) + 1.00784 (Hydrogen Atomic Mass)
Butyric acid acetate	6.57%	146.048
Dehydroascorbic acid	8.92%	173.095
D-Glucopyranose	5.08%	179.058
D-Gluconic acid	6.45%	195.053
Hydroxycitric acid	4.25%	207.09
Phenyl-D-galactopyranoside	22.91%	255.235
6-O-(4-Hydroxy-2-methylenebutanoyl)-β-D-glucopyranose	18.3%	277.220
Flavones	3.57%	278.225
Medicagol	4.76%	295.231
Mellitic acid	10.8%	341.112
Trehalose	24.84%	377.089
Pelentanic acid	8.35%	379.087
Kaempferol 3-O-arabinoside	11.82%	417.108
Apigenin 7-O-(6-malonyl-β-D-glucoside)	6.57%	517.182
Galactopyranosyl-D-glucose hydrate	36.67%	701.516
6-O-β-D-Glucopyranosyl-1-O-[(2E)-3-(4-hydroxy-3,5-dimethoxyphenyl)-2-propenoyl]-2-O-[(2E)-3-(4-hydroxy-3-methoxyphenyl)-2-propenoyl]-β-D-glucopyranose	83.97%	723.501
5,7-Dihydroxy-2-(4-hydroxyphenyl)-4-oxo-4H-chromen-3-yl 6-deoxy-α-L-mannopyranosyl-(1->3)-[β-D-xylopyranosyl-(1->2)]-α-D-glucopyranoside.	100%	725.516

## Data Availability

Original data and the product extract tested are available under the responsibility of Prof. Domenico Tricarico.

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
