# Peer review of "Molecular Composition and Biological Activity of a Novel Acetonitrile–Water Extract of Lens Culinaris Medik in Murine Native Cells and Cell Lines Exposed to Different Chemotherapeutics Using Mass Spectrometry"

_cells, 2023, doi:10.3390/cells12040575_

Round 1

Reviewer 1 Report

Despite of the novel developed formulation is improved in terms of the observed effects in respect to the previously published  hydroalcoholic extract from the same labs, some questions remain open, for instance the authors proposed some candidates in the osteoblast and bone marrow cells protection against very potent cytotoxic compounds such as doxorubucine and staurosporine and in the renal and neuronal protections but no quantification of the individual molecules were provided. This need to be explained or at least justified in a specific paragraph in the discussion.

Mayor concern:

Abstract :

Line 34 please delete  “helpful in sarcoma” hypothesis not tested in the present work 

Results:

-figures 6, 7 and 8 can be combined in one figure

-high resolution figures must be provided.

Discussion :

-can be shorten focusing on the main findings and conclusions

-a paragraph reporting the limitation of their work must be provided in the discussion sections

Minor concerns:

English Editing is required

Line 665: replace extract with  solvent

Line 678-735 shoul be more concise

Line 739: check for redundancy

Revised reference n° 38

Author Response

We thank the reviewer for his positive comment on our manuscript. 

Mayor concern:

Abstract :

Line 34 Please delete  “helpful in sarcoma” hypothesis not tested in the present work 

ok

Results:

-figures 6, 7, and 8 can be combined into one figure

-high-resolution figures must be provided.

ok

Discussion :

-can be shorten focusing on the main findings and conclusions

ok

-a paragraph reporting the limitation of their work must be provided in the discussion sections

ok

Minor concerns:

English Editing is required

ok

Line 665: replace extract with  solvent

ok

Line 678-735 should be more concise

ok

Line 739: Check for redundancy

ok

Revised reference n° 38

ok

Reviewer 2 Report

In figure 9a, change the plot and/or brown color because the groups are not differentiated

For figure 9b the pink groups do not differ from each other

In figure 10a use a different dither or color because it is difficult to distinguish the different groups of green color

Author Response

We thank the reviewer for her/his positive comment on our manuscript. 

Comments and Suggestions for Authors

In figure 9a, change the plot and/or brown color because the groups are not differentiated

For figure 9b the pink groups do not differ from each other

In figure 10a use a different dither or color because it is difficult to distinguish the different groups of green color

Answer

As requested Figures 9ab and 10ab were revised to better differentiate between groups